A DNA barcode survey of insect biodiversity in Pakistan

Ashfaq Muhammad 1 mashfaq@uoguelph.ca
Khan Arif M. 2
Rasool Akhtar 3
Akhtar Saleem 4
Nazir Naila 5
Ahmed Nazeer 6
Manzoor Farkhanda 7
Sones Jayme 8
Perez Kate 8
Sarwar Ghulam 9
Khan Azhar A. 10
Akhter Muhammad 11
Saeed Shafqat 12
Sultana Riffat 13
Tahir Hafiz Muhammad 14
Rafi Muhammad A. 15
Iftikhar Romana 16
Naseem Muhammad Tayyib 17
Masood Mariyam 18
Tufail Muhammad 19
Kumar Santosh 20
Afzal Sabila 21
McKeown Jaclyn 8
Samejo Ahmed Ali 13
Khaliq Imran 19
D’Souza Michelle L. 8
Mansoor Shahid 22
http://orcid.org/0000-0002-3081-6700 Hebert Paul D. N. 1
1 Centre for Biodiversity Genomics & Department of Integrative Biology, University of Guelph , Guelph , Canada
2 Department of Biotechnology, University of Sargodha , Sargodha , Pakistan
3 Centre for Animal Sciences and Fisheries, University of Swat , Mingora , Pakistan
4 Directorate of Entomology, Ayub Agricultural Research Institute , Faisalabad , Pakistan
5 Department of Entomology, University of Poonch , Rawalakot, Azad Kashmir , Pakistan
6 Faculty of Life Sciences and Informatics, Balochistan University of Information Technology, Engineering and Management Sciences , Quetta , Pakistan
7 Department of Zoology, Lahore College for Women University , Lahore , Pakistan
8 Centre for Biodiversity Genomics, University of Guelph , Guelph , Canada
9 Institute of Zoology, University of the Punjab , Lahore , Pakistan
10 College of Agriculture, Bahauddin Zakariya University Bahadur Campus , Layyah , Pakistan
11 Pulses Research Institute, Ayub Agricultural Research Institute , Faisalabad , Pakistan
12 Faculty of Agriculture and Environmental Sciences, MNS University of Agriculture , Multan , Pakistan
13 Department of Zoology, University of Sindh , Jamshoro , Pakistan
14 Department of Zoology, Government College University Lahore , Lahore , Pakistan
15 National Insect Museum, National Agricultural Research Center , Islamabad , Pakistan
16 Department of Plant Pathology, Washington State University , Pullman, WA , United States
17 Department of Biology, University of Copenhagen , Copenhagen , Denmark
18 Government College Women University Faisalabad , Faisalabad , Pakistan
19 Ghazi University , Dera Ghazi Khan , Pakistan
20 Department of Zoology, Cholistan University of Veterinary and Animal Sciences , Bahawalpur , Pakistan
21 Department of Zoology, University of Narowal , Narowal , Pakistan
22 National Institute for Biotechnology and Genetic Engineering , Faisalabad , Pakistan
Gillespie Joseph
Electronic publication date: 2022 Apr 25
Publication date: 2022
Volume: 10
Electronic Location ID: e13267
Received 2021 Dec 30; Accepted 2022 Mar 23
Copyright: © 2022 Ashfaq et al.
Copyright year: 2022
Copyright holder: Ashfaq et al.
License: This is an open access article distributed under the terms of the Creative Commons Attribution License, which permits unrestricted use, distribution, reproduction and adaptation in any medium and for any purpose provided that it is properly attributed. For attribution, the original author(s), title, publication source (PeerJ) and either DOI or URL of the article must be cited.
License URL: https://creativecommons.org/licenses/by/4.0/

Keywords: DNA barcoding, Cytochrome c oxidase I, Barcode index number, Biodiversity overlap, BOLD

Funding: International Development Research Centre (IDRC) in Canada 106106-001 Higher Education Commission of Pakistan 20-1403/R& D/09 Government of Canada Canada First Research Excellence Fund New Frontiers in Research Fund This study was enabled by grant 106106-001 “Engaging Developing Nations in iBOL” from the International Development Research Centre (IDRC) in Canada and by grant HEC No. 20-1403/R& D/09 “Sequencing DNA Barcodes of Economically Important Insect Species from Pakistan” from the Higher Education Commission of Pakistan. Sequence analysis was made possible by a grant from the Government of Canada through Genome Canada and Ontario Genomics in support of the International Barcode of Life (iBOL) project. This is a contribution to the ‘Food from Thought’ project supported by the Canada First Research Excellence Fund. Paul Hebert and Nazeer Ahmed were supported by the New Frontiers in Research Fund through its award to BIOSCAN. The funders had no role in study design, data collection and analysis, decision to publish, or preparation of the manuscript.

==============================
Although Pakistan has rich biodiversity, many groups are poorly known, particularly insects. To address this gap, we employed DNA barcoding to survey its insect diversity. Specimens obtained through diverse collecting methods at 1,858 sites across Pakistan from 2010–2019 were examined for sequence variation in the 658 bp barcode region of the cytochrome c oxidase 1 (COI) gene. Sequences from nearly 49,000 specimens were assigned to 6,590 Barcode Index Numbers (BINs), a proxy for species, and most (88%) also possessed a representative image on the Barcode of Life Data System (BOLD). By coupling morphological inspections with barcode matches on BOLD, every BIN was assigned to an order (19) and most (99.8%) were placed to a family (362). However, just 40% of the BINs were assigned to a genus (1,375) and 21% to a species (1,364). Five orders (Coleoptera, Diptera, Hemiptera, Hymenoptera, Lepidoptera) accounted for 92% of the specimens and BINs. More than half of the BINs (59%) are so far only known from Pakistan, but others have also been reported from Bangladesh (13%), India (12%), and China (8%). Representing the first DNA barcode survey of the insect fauna in any South Asian country, this study provides the foundation for a complete inventory of the insect fauna in Pakistan while also contributing to the global DNA barcode reference library.

Introduction

With an area of 882,000 km2, Pakistan includes seven biomes (Deserts & Xeric Shrublands, Flooded Grasslands & Savannas, Mangroves, Montane Grasslands & Shrublands, Temperate Broadleaf & Mixed Forests, Temperate Conifer Forests, Tropical & Subtropical Coniferous Forests) and portions of the Indo-Malayan and Palearctic biogeographic realms (Dinerstein et al., 2017). Because of this physiographic and climatic variation, its faunal diversity is quite high (Anwar, Jasra & Ahmad, 2008). While its vertebrate fauna is well known (Khan, 1976; Roberts, 1997; Grimmett, Roberts & Inskipp, 2008), prior studies on other animal lineages have been restricted to specific taxa or regions (Inayat et al., 2010; Iftikhar et al., 2016a; Manzoor, Khan & Shah, 2020). Although over 5,000 insect species have been reported (Government of Pakistan, 2000), no taxonomically comprehensive or wide-ranging assessments have been undertaken due to the scarcity of taxonomists, and the fact that many insect species remain undescribed (Ward & Lariviere, 2004). Because conventional morphological approaches (Pompeo et al., 2017) are difficult to implement at scale (Fattorini, 2013), the species count for Pakistan remains uncertain as it does globally (Scheffers et al., 2012).

The effectiveness of DNA barcoding (Hebert et al., 2003) in both specimen identification and species discovery (Hebert et al., 2004; Huemer et al., 2014; Kress et al., 2015) has stimulated its rapid adaptation in modern biodiversity studies (Ashfaq & Hebert, 2016; DeSalle & Goldstein, 2019). This work has generated DNA barcode coverage for more than 760,000 animal species on the Barcode of Life Data System (BOLD) (www.boldsystems.org) (Ratnasingham & Hebert, 2007). The effectiveness of the Barcode Index Number (BIN) system (Ratnasingham & Hebert, 2013) as a species proxy (Hausmann et al., 2013) has made it possible to rapidly evaluate species diversity, enabling large-scale biotic inventories (Hebert et al., 2016; Wirta et al., 2016). Because BINs show close congruence with species boundaries established through morphological study (Ortiz et al., 2017; Huemer et al., 2019) they can be used to delineate newly encountered species (Mitchell, Moeseneder & Hutchinson, 2020), to discern cryptic species (Zahiri et al., 2017; Zhou et al., 2019), to plot species distributions (Ren et al., 2018), to estimate species richness in bulk samples (Andújar et al., 2018; Braukmann et al., 2019), to analyze museum collections (Pentinsaari et al., 2020), and to assess faunal similarity at regional and global scales (Ashfaq et al., 2017).

The effectiveness of DNA barcoding coupled with advances in sequencing technology allow it to support large-scale biodiversity analysis (Wilson et al., 2017). However, the intensity of study has varied among regions (Weigand et al., 2019). For example, the BIN count (84,000) for Canada is 8× that for Russia (11,000), although the latter nation is 1.7× larger (www.boldsystems.org, accessed 7 September 2021). In a similar fashion, the BIN count for Germany (23,000) is 4× that for India (5,800), although the latter nation is 9× larger. The current study extends DNA barcode coverage for Pakistan to both advance knowledge of the taxonomic composition of its insect fauna and to develop a barcode reference library that supports routine eDNA and metabarcoding studies in the future.

Materials and Methods

Sample collection and preparation

Insects were sampled at 1,858 sites across Pakistan (Fig. 1) from 2010–2019 using both active and passive collecting methods including sweep nets, hand collections, hostplant beating, light traps, Malaise traps, pitfall traps, and UV illuminated sheets. Plans for the specimen collections/sites were approved by the Director, National Institute for Biotechnology and Genetic Engineering, Faisalabad under the project HEC No. 20-1403/R& D/09. The specimens were identified to an order and, where possible, to lower taxonomic ranks. Large specimens were either pinned and preserved dry or placed in Matrix tubes with 95% ethanol. Small specimens were individually placed in a well containing 30 μl of 95% ethanol in 96-well microplates. Specimen metadata and an image (except for Malaise samples where only representative specimens of each BIN were imaged) were submitted to BOLD where the information can be accessed on both the specimen page and corresponding BIN page. Voucher specimens are archived at the National Institute for Biotechnology and Genetic Engineering (NIBGE), Faisalabad, Pakistan (with sample ID prefix NIBGE) or at the Centre for Biodiversity Genomics (CBG), Guelph, Canada (with ID prefix BIOUG).

Figure 1 Map showing the collection sites for the insects examined in this study. The size and color of each site point indicate the number of specimens sampled. Map was generated in R using Google Maps satellite imagery.

DNA barcoding

A total of 60,273 insects were barcoded following standard protocols (deWaard et al., 2019a, 2019b). In brief, a leg was removed with sterile forceps from each large specimen and transferred to a well preloaded with 30 μl of 95% EtOH. As smaller specimens were already in plates, they were ready for analysis, but vouchers were recovered after DNA extraction (Porco et al., 2010). DNA extraction, PCR amplification, and sequencing were performed at the Canadian Centre for DNA Barcoding (CCDB) following established protocols (Ivanova, deWaard & Hebert, 2006; Hebert et al., 2018; deWaard et al., 2019b). PCR reactions were either 6 μl or 12 μl (Hebert et al., 2013). Three quarters (73%) of the specimens were Sanger sequenced while the rest were analyzed using SMRT sequencing on a Sequel platform (Pacific Biosciences, Menlo Park, CA, USA). Sanger sequencing employed BigDye Terminator Cycle Sequencing Kit (v3.1) on an Applied Biosystems 3730XL DNA Analyzer. Sequences were assembled, aligned and edited using CodonCode Aligner before submission to BOLD. SMRT sequencing employed protocols described by Hebert et al. (2018). The resultant sequences were uploaded to mBRAVE (Multiplex Barcoding Research and Visualization Environment; www.mbrave.net) for editing (sequence trimming, quality filtering, de-replication), identification, and generation of operational taxonomic units (OTUs). The edited sequences were subsequently exported to BOLD for BIN assignment and reference library development. The specimen records, sequence data, electropherograms, and primer details are available in the dataset “DS-INSCTPAK” (dx.doi.org/10.5883/DS-INSCTPAK). All DNA extracts are stored within the DNA archive facility at the CBG.

Data analysis

The final dataset (N = 50,592) included 50,094 new barcode records and 498 public records on BOLD from specimens collected in Pakistan (Table S1). All records were assigned taxonomy and BINs following the workflow outlined by deWaard et al. (2019b). In brief, once the barcode data was on BOLD, each record went through a taxonomic assignment and verification workflow. Earlier studies (Ashfaq et al., 2013; Nazir et al., 2014; Iftikhar et al., 2016b; Akhtar et al., 2018; Naseem et al., 2019) on five taxa (antlions, aphids, butterflies, grasshoppers, thrips) coupled analysis of barcode results with detailed morphological study by taxonomic specialists. All sequences meeting the quality criteria were either assigned to an existing BIN or founded a new one (Ratnasingham & Hebert, 2013). Sequences founding a new BIN had to possess >500 bp of the barcode region with <1% ambiguous bases and no stop codons. Shorter sequences (300–495) that met the latter two quality criteria and that were a close sequence match to an established BIN were assigned to it (deWaard et al., 2019a). The remaining short sequences (1,230) that failed to gain a BIN assignment were run through the stand-alone version of the RESL algorithm (Ratnasingham & Hebert, 2013) (using the function Cluster Sequences on BOLD) to estimate the number of additional OTUs among them. One representative from each OTU was then queried against the BOLD ID Engine to link them with known BINs (deWaard et al., 2019b). The BIN details with specimen records and representative images (where available) are accessible on BOLD (dx.doi.org/10.5883/DS-INSCTPAK).

Various statistical approaches were used to estimate the number of insect species in Pakistan (Chao & Chiu, 2016) including the parametric estimator Preston’s log-normal as well as non-parametric estimators Chao1, and the first-order and second-order jackknife. A bias-corrected version of each non-parametric estimator, designed to improve performance under conditions of low sampling effort, was also included (Lopez et al., 2012). All estimates were calculated using the R packages vegan and BAT. In addition, a species accumulation curve was drawn based on a sample-size-based rarefaction and extrapolation to at most double the minimum observed sample size, guided by an estimated asymptote using the R package iNEXT (Hsieh, Ma & Chao, 2016).

Results

DNA barcodes were recovered from 50,094 (83%) of the 60,273 specimens analyzed. The other 17% either failed to amplify or generated problematic sequences (e.g., contamination, NUMTs, stop codons, endosymbionts) that were excluded from subsequent analysis. Considering orders with 100 or more specimens, sequence recovery ranged from a low of 63% for Blattodea to 95% for Lepidoptera. Sequence recovery for the other four major orders of insects showed considerable variation (Diptera: 91%, Coleoptera: 80%, Hymenoptera: 78%, Hemiptera: 69%).

All 50,592 insects with a barcode were assigned to one of 19 orders while 99.8% received an assignment to one of 362 families (Table 1, Table S1). Five orders represented 92% of the specimens: Diptera (40%), Hymenoptera (21%), Lepidoptera (12%), Hemiptera (11%), and Coleoptera (8%) (Fig. 2). Six orders (Mantodea, Neuroptera, Odonata, Orthoptera, Psocodea, Thysanoptera) were each represented by >100 specimens while the remaining eight possessed fewer representatives (Fig. 2, Table S1). Most of these sequences (98%) received a BIN assignment, leading to a total of 6,590 BINs. The other 1,230 barcode sequences did not meet the criteria for BIN assignment but included 629 OTUs when analyzed using “Cluster Sequences” function on BOLD. The BOLD ID Engine assigned 82 of these OTUs to known BINs, but the other 547 OTUs likely represent taxa new to BOLD. Many (57%) of the 6,590 BINs were represented by two or more sequences, but 43% were represented by just a single specimen. The ratio of these singletons was above 40% in all five major orders but was highest in Hymenoptera (48%). Most BINs (88%; N = 5,754) possessed an image of at least one voucher.

Figure 2 Pie chart showing the number of specimens barcoded from each of the 19 insect orders. Different colors represent different orders. Numbers next to each slice indicate the specimen count for the order.

Table 1 Number of specimens belonging to 19 insect orders from Pakistan with DNA barcode records. The number of families, genera, species, and BINs is reported for each order.

Order	Specimens with barcodes	Specimens assigned to BINs (%)	BINs recovered	OTUs without
BIN*	Singleton BINs (%)	BINs assigned to family (%)	Families
recovered	BINs assigned to genus (%)	Genera recovered	BINs assigned to species (%)	Species recovered	
Blattodea	64	84	19	5	36.8	100	5	78.9	9	52.6	10	
Coleoptera	3,889	93	819	123	45.2	100	56	21.9	118	13.3	119	
Dermaptera	24	83.3	3	2	33.3	100	2	33.3	1	0.0	0	
Diptera	20,095	99	1,684	94	40.1	99.0	68	29.8	212	13.8	222	
Embioptera	28	96.4	7	1	14.3	100	2	14.3	1	14.3	1	
Hemiptera	5,859	96.5	642	73	41.9	98.3	59	31.6	132	22.6	135	
Hymenoptera	10,542	96	1,711	177	47.7	99.4	50	34.7	226	10.2	170	
Lepidoptera	6,064	99.4	1,233	24	42.5	99.6	62	71.9	516	41.5	514	
Mantodea	113	97.3	36	2	50.0	100	2	13.9	4	5.6	2	
Megaloptera	6	100	1	0	0.0	100	1	100	1	100	1	
Neuroptera	559	92.3	99	6	39.4	99.0	7	54.5	30	36.4	32	
Odonata	353	92.6	51	11	21.6	100	12	92.2	30	88.2	47	
Orthoptera	1,409	97.59	163	21	30.1	100	12	44.2	53	37.4	54	
Phasmatodea	4	75	3	1	100.0	100	1	0.0	0	0.0	0	
Psocodea	950	97.5	31	5	22.6	93.5	13	38.7	10	19.4	6	
Strepsiptera	2	100	1	0	0.0	100	1	100	1	0.0	0	
Thysanoptera	618	99.3	76	2	34.2	100	3	80.3	27	69.7	48	
Trichoptera	11	100	10	0	90.0	100	6	60.0	4	40.0	4	
Zygentoma	2	100	1	0	0.0	0.0	0	0.0	0	0.0	0	
Total	50,592	97.6%	6,590	547	42.9%	99%	362	40%	1,375	21%	1,364	
Note:

* For recognition as a new BIN, a sequence must include >500 bp of the barcode region (positions 70 bp to 700 bp in the BOLD alignment) and possess <1% ambiguous bases.

The percentage of records in each of the five major orders with a BIN assignment ranged from 93% (Coleoptera) to 99% (Diptera, Lepidoptera) with Hemiptera and Hymenoptera intermediate (96%) (Table 1). These five orders also contributed most of the BINs (92%) and families (81%) (Table 1, Figs. 3A, 3B). Only 40% of BINs were placed to a genus and 21% to a species, but this still led to records for 1,375 genera and 1,364 species (Table 1, Table S1). Among the five major orders, more BINs were identified to a genus (72%) and species (41%) in Lepidoptera than in the other four orders (Table 1). For example, just 13.8% of Diptera BINs and 10.2% of Hymenoptera BINs were assigned to a species (Table 1).

Figure 3 Taxonomic (A) and BIN assignments (B) for the 12 insect orders represented by >50 specimens. Species assignment in (A) is based on the assignment of barcode(s) to the named species on BOLD.

Specimen counts for the 362 families were highly variable as 15 families were each represented by >1,000 specimens while 38 had just one (Table S1). This pattern was also reflected in the number of BINs as 15 families had >100 BINs while 86 had just one. The Chironomidae (N = 3,258) and Braconidae (N = 2,174) were represented by the most specimens while Cecidomyiidae (238 BINs) and Platygastridae (230 BINs) were most diverse. Figure 4 shows the BIN diversity and BIN:specimen ratio for the 15 families with >100 BINs. The ratio was highest (0.33) for Geometridae (Lepidoptera) and lowest (0.05) for Chironomidae. The species accumulation curve did not reach an asymptote indicating more species await detection (Fig. 5). Species estimates for the country ranged from 9,253 to 12,246 species suggesting that, on average, 40% of species remain to be sampled (Table 2).

Figure 4 BIN diversity and BIN: specimen ratio for the 15 insect families represented by >100 BINs.

Figure 5 Sample-size-based rarefaction (solid line) and extrapolation (dashed line) sampling curves for 49,363 specimens with barcodes from Pakistan. Solid dots represent the observed richness of 6,590 species. The curve is estimated to reach an asymptote at 10,382 species.

Table 2 Species richness estimates based on the abundances of the 6,590 insect BINS encountered at 1,858 sites across Pakistan.

SPECIMENS	BINS	PRESTON	CHAO1	CHAO1P	JACK1AB	JACK1ABP	JACK2AB	JACK2ABP	
49,363	6,590	9,253	10,377	12,285	9,416	11,147	11,189	12,246	
Note:

Seven estimates were calculated: Preston’s log-normal (PRESTON), Chao1 (CHAO1), first-order jackknife (JACK1AB), second-order jackknife (JACK2AB), and their bias-corrected complements (CHAO1P, JACK1ABP, JACK2ABP).

BOLD was searched to ascertain if the 6,590 insect BINs from Pakistan were known from other countries. This analysis showed that 2,684 BINs (41%) were shared with at least one of 199 other countries while the others (3,906) are so far only known from Pakistan. The percentage of shared species ranged from 0.02% to 13%. Figure 6 shows the overlap values between Pakistan and countries with >1,000 BINs. BIN overlap was higher with nearby countries (Bangladesh: 13%, India: 12%, China: 8%) than for other regions. For example, Pakistan shared just 5% of its BINs with Australia, South Africa, and Germany (Fig. 6). The overlap between Canada and Costa Rica, both countries with >50,000 insect BINs, was only 4% and 1% respectively (Fig. 6).

Figure 6 Percentage of insect BINs shared between Pakistan and the 70 other nations with >1,000 insect BINs on the Barcode of Life Data Systems (BOLD).

Discussion

Current estimates of the number of insect species which occur in Pakistan range from 5,000 (Ministry of Climate Change, Pakistan, 2019) to 20,000 species (Hasnain, 1998), but they are certainly too low (Baig & Al-Subaiee, 2009). The current study aimed to refine estimates of species richness by coupling DNA barcoding with the BIN system. With over 50,000 specimens sequenced, this study represents, by far, the largest effort to assemble a DNA barcode registry for the insect fauna of any South Asian country. While success (83%) in DNA barcode recovery was good, it varied considerably among orders from 63% for Blattodea to 95% for Lepidoptera. Similar variation in barcode recovery among different insect taxa has been reported in other studies (Geiger et al., 2016; Pentinsaari et al., 2020). For example, a study on the insect fauna of French Polynesia reported 91% recovery for Diptera vs 63% for Coleoptera (Ramage et al., 2017). Similarly, a large-scale Canadian study revealed 95% recovery for Diptera vs 77% for Hemiptera and 74% for Hymenoptera (deWaard et al., 2019a). Although DNA quantity and quality play an important role (Ballare et al., 2019; Velasco-Cuervo et al., 2019), failures in primer binding often underlie low sequence recovery (Hajibabaei et al., 2005, Hebert et al., 2016). Such failures can lead to the underestimation of species richness in insect groups where recovery is low (Hebert et al., 2016). Other factors, such as co-amplification of pseudogenes (Song et al., 2008), Wolbachia (Smith et al., 2012), recent speciation (van Velzen et al., 2012), or incomplete lineage sorting (Mallo & Posada, 2016) may also limit the efficacy of barcodes to delimitate species, consequently affecting the diversity estimates. Moreover, there are instances where the BIN system overestimated species diversity in certain insect groups, such as Chironomidae (Lin, Stur & Ekrem, 2015; Ekrem et al., 2018).

The coupling of morphological inspection with barcode matches on BOLD (deWaard et al., 2019a, 2019b) was very effective at placing BINs to an order (100%) and family (>99%). However, just 40% of the BINs could be assigned to a genus and 21% to a species indicating the need for better parameterization of the barcode reference library. This was particularly true for the three most diverse orders where species assignments were less than 15% (Diptera: 13.8%, Coleoptera: 13.3%, Hymenoptera: 10.2%). Considerably higher assignment success has been reported for Malaise samples from Germany (34%) and Canada (38%) (Geiger et al., 2016; deWaard et al., 2019a) reflecting the more comprehensive DNA barcode reference libraries available for these nations. Despite the limited reference database (Virgilio et al., 2010), the present analysis identified representatives from 1,375 genera and 1,364 species showing the value of the global reference library (BOLD) which far exceeds the results obtained by morphology alone (Marshall, Paiero & Buck, 2009). The present analysis revealed 6,590 BINs with species richness estimates indicating that the fauna of Pakistan certainly includes more than 10,000 species. As these estimates are based on specimens collected with uneven sampling and limited geographic coverage, they are likely to increase with more comprehensive efforts.

Although 19 insect orders were detected, five (Coleoptera, Diptera, Hemiptera, Hymenoptera, Lepidoptera) were dominant (92%), reinforcing prior results from morphological (Stork, 2018) and barcoding studies (Ritter et al., 2019; Pentinsaari et al., 2020). Malaise traps preferentially capture low-flying insects such as Diptera and Hymenoptera (Cooksey & Barton, 1981; deWaard et al., 2019b), the two orders that made 61% of the collections. Other studies have reported a similar pattern (Brown, 2005; Karlsson et al., 2020). For example, a Canadian study found that Diptera comprised 57% of the collections (deWaard et al., 2019b).

Fifteen of the 362 families dominated with 1,000 or more specimens and this pattern was also reflected in the BIN diversity. The uneven detection of families in the survey is supported by the fact that 38 families were represented by just one specimen and 88 by one BIN. Interestingly, nine of the 15 families with the most BINs were dipterans and hymenopterans with Cecidomyiidae and Ichneumonidae comprising the highest BIN:specimen ratio.

Because BOLD now hosts around nine million DNA barcode records for more than 760,000 animal species, it provides a good basis for assessing faunal overlap using BINs. Only 41% of the 6,590 insect BINs from Pakistan are currently known from other countries. As expected, BIN overlap was highest with neighboring countries. This result reflects the endemism of biodiversity (Werneck et al., 2012) and underscores the need to develop local biodiversity inventories. The current survey represents a first step towards building an inventory for the insect fauna of Pakistan.

Supplemental Information

Supplemental Information 1 BIN and taxonomic assignments of 50,592 specimens analyzed in the study.

Click here for additional data file.

Additional Information and Declarations

Competing Interests

Author Contributions

Field Study Permissions

Data Availability

The authors declare that they have no competing interests.

Muhammad Ashfaq conceived and designed the experiments, performed the experiments, analyzed the data, prepared figures and/or tables, authored or reviewed drafts of the paper, sample collections, and approved the final draft.

Arif M. Khan conceived and designed the experiments, performed the experiments, prepared figures and/or tables, authored or reviewed drafts of the paper, sample collections, and approved the final draft.

Akhtar Rasool conceived and designed the experiments, performed the experiments, prepared figures and/or tables, authored or reviewed drafts of the paper, sample collections, and approved the final draft.

Saleem Akhtar conceived and designed the experiments, performed the experiments, prepared figures and/or tables, authored or reviewed drafts of the paper, sample collections, and approved the final draft.

Naila Nazir conceived and designed the experiments, performed the experiments, analyzed the data, prepared figures and/or tables, authored or reviewed drafts of the paper, sample collections, and approved the final draft.

Nazeer Ahmed conceived and designed the experiments, performed the experiments, analyzed the data, prepared figures and/or tables, sample collections, and approved the final draft.

Farkhanda Manzoor conceived and designed the experiments, performed the experiments, analyzed the data, prepared figures and/or tables, sample collections, and approved the final draft.

Jayme Sones conceived and designed the experiments, performed the experiments, analyzed the data, authored or reviewed drafts of the paper, and approved the final draft.

Kate Perez conceived and designed the experiments, performed the experiments, analyzed the data, authored or reviewed drafts of the paper, and approved the final draft.

Ghulam Sarwar conceived and designed the experiments, performed the experiments, analyzed the data, authored or reviewed drafts of the paper, sample collections, and approved the final draft.

Azhar A. Khan conceived and designed the experiments, performed the experiments, authored or reviewed drafts of the paper, sample collections, and approved the final draft.

Muhammad Akhter conceived and designed the experiments, performed the experiments, authored or reviewed drafts of the paper, sample collections, and approved the final draft.

Shafqat Saeed conceived and designed the experiments, performed the experiments, analyzed the data, authored or reviewed drafts of the paper, specimen identification, and approved the final draft.

Riffat Sultana conceived and designed the experiments, performed the experiments, analyzed the data, prepared figures and/or tables, authored or reviewed drafts of the paper, sample collections, and approved the final draft.

Hafiz Muhammad Tahir conceived and designed the experiments, performed the experiments, analyzed the data, authored or reviewed drafts of the paper, specimen identification, and approved the final draft.

Muhammad A. Rafi conceived and designed the experiments, performed the experiments, analyzed the data, authored or reviewed drafts of the paper, specimen identification, and approved the final draft.

Romana Iftikhar conceived and designed the experiments, performed the experiments, analyzed the data, prepared figures and/or tables, authored or reviewed drafts of the paper, sample collections, and approved the final draft.

Muhammad Tayyib Naseem conceived and designed the experiments, performed the experiments, analyzed the data, prepared figures and/or tables, authored or reviewed drafts of the paper, sample collections, and approved the final draft.

Mariyam Masood conceived and designed the experiments, performed the experiments, analyzed the data, authored or reviewed drafts of the paper, sample collections, and approved the final draft.

Muhammad Tufail conceived and designed the experiments, performed the experiments, authored or reviewed drafts of the paper, sample collections, and approved the final draft.

Santosh Kumar conceived and designed the experiments, performed the experiments, authored or reviewed drafts of the paper, sample collections, and approved the final draft.

Sabila Afzal conceived and designed the experiments, performed the experiments, authored or reviewed drafts of the paper, sample collections, and approved the final draft.

Jaclyn McKeown performed the experiments, analyzed the data, authored or reviewed drafts of the paper, specimen imaging, and approved the final draft.

Ahmed Ali Samejo conceived and designed the experiments, performed the experiments, analyzed the data, authored or reviewed drafts of the paper, sample collections, and approved the final draft.

Imran Khaliq conceived and designed the experiments, performed the experiments, authored or reviewed drafts of the paper, sample collections, and approved the final draft.

Michelle L. D’Souza analyzed the data, prepared figures and/or tables, authored or reviewed drafts of the paper, and approved the final draft.

Shahid Mansoor conceived and designed the experiments, performed the experiments, authored or reviewed drafts of the paper, sample collections, and approved the final draft.

Paul D. N. Hebert conceived and designed the experiments, performed the experiments, analyzed the data, prepared figures and/or tables, authored or reviewed drafts of the paper, specimen identifications, and approved the final draft.

The following information was supplied relating to field study approvals (i.e., approving body and any reference numbers):

Field experiments were approved by the Director National Institute for Biotechnology and Genetic Engineering Faisalabad (project number; HEC No. 20-1403/R& D/09).

The following information was supplied regarding data availability:

The sequences are available at Barcode of Life Data Systems: dx.doi.org/10.5883/DS-INSCTPAK.

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
