# Peer review of "A DNA barcode survey of insect biodiversity in Pakistan"

_PeerJ, doi:10.7717/peerj.13267_

## Round 0.1 · original submission · Major Revisions

Dear Dr. Ashfaq and colleagues:

Thanks for submitting your manuscript to PeerJ. I have now received three independent reviews of your work, and as you will see, one reviewer recommended rejection, while the others suggested minor revisions (though with many suggested changes). I am affording you the option of revising your manuscript according to all reviews but understand that your resubmission may be sent to at least one new reviewer for a fresh assessment (unless the reviewer recommending rejection is willing to re-review).

Some concern was raised about the relevance of barcoding unidentified insects. For instance, metabarcoding Malaise trap samples could unveil some interesting questions about local diversity and variation across samples, but that was not carried out in your work. Your study only reported on the number of matches in BINs and proportion of matches with BoLD identified species. This has severe limitations given that the library is not propagated with curated sequences, leaving the majority of specimens unidentified. If the purpose would be to one day enable the detection of a species that are identified with a barcode in the future, this should be explained with a detailed approach.

Please note that Reviewer 1 kindly provided a marked-up version of your manuscript.

There are many minor suggestions to improve the manuscript. Importantly, please ensure that an English expert has edited your revised manuscript for content and clarity.

Therefore, I am recommending that you revise your manuscript, accordingly, taking into account all of the issues raised by the reviewers.

Good luck with your revision,

-joe

·

Basic reporting

In this study Ashfaq provide the first DNA barcode library for insects for a South Asian country. The library includes almost 49,000 DNA barcodes that were uploaded to BOLD and assigned to 6,590 Barcode Index Numbers. This study bears evidence for a long and comprehensive, careful work. Finally, it is another proof of that DNA barcoding represents an effective molecular method to identify specimens. Without doubt, the topic of this manuscript is interesting and appropriate for “PeerJ”.

Experimental design

The research question is well explained, the used methods represent state-of-the-at approaches to analyze DNA barcode data, using the BOLD workbench and offered tools. All data are open to the public and not deposited in local data bases.

Validity of the findings

As already pointed out, I think that the topic of this manuscript is very interesting and appropriate for “PeerJ”. It represents an important step in building-up comprehensive DNA barcode libraries for Asia which. Such libraries represent the backbone of any modern biodiversity studies using high-throughput sequencing technologies (e.g., “meta-barcoding”, eDNA analysis). Furthermore, it highlights the necessity of on-going taxonomic studies and efforts to document the insect species diversity of Pakistan. It is hoped that the present sequence data will become the basis for new species descriptions.

Additional comments

If possible, it would be nice to add some images of selected insect species in order to document the enormous diversity of the insect fauna of Pakistan.

Furthermore, please check the uploaded pdf for additional comments.

Reviewer 2 ·

Basic reporting

The manuscript is well written without any obvious grammatical or other language problems. Many relevant references are cited.
Figures:
Figure 2: colours not possible to interpret - please add taxon names to the actual pies
Figure 3: these simple Excel figures are fairly ok, but not particularly elegant. In (a), what is a 'species' - as identified by BoLD search?
Figure 6: better to rank countries by their insect faunal similarity

Raw data are available in BoLD.

It is not immediately clear how the hypothesis on Pakistan diversity can be tested with massive DNA barcoding.

Experimental design

For the purpose of the study the experimental design is fine. Data obtained are trusted and handled correctly and possible to inspect in BoLD. Sample size is very large.

There are some questions of interest being tested and answered such as how many more samples are needed to cover the near entire fauna of insects (rarefaction and similar measures).

The main problem with this work is the obviousness of the data. Yes, the main holometabolan orders were the most prevalent in the sampled specimens; yes, the diversity is underestimated, but estimates are fairly close to the tallied numbers from the literature; yes, the BoLD (and mirrored GenBank) databases are not densely populated for the Pakistani fauna; yes, many bins have no names (because taxonomists are not invited or take years to get an answer); yes, more bins are related to neighboring countries than far away, even though there are a certain proportion of widespread species, often pest species.

So then - the 1000$ question - what is the point of doing all the work? It certainly does not provide any new information about the Pakistani fauna! The BoLD (and GenBank) database is propagated with sequences from unidentified specimens. In most cases these data do not provide any useful information and it will clog the system in the end. A kind advice for the future is to focus on fewer target groups, and connect with good taxonomists long before the study is initiated. Then you will get highly informative data, including necessary corrections to the current taxonomy and at the same time you will get estimates on diversity and sampling properties.

Validity of the findings

See above. Impact and novelty marginal, but data are clean, and properly curated.

Additional comments

Line 224: coupling of morphological inspection with barcode matches - how exactly did it facilitate the provision of BINs as stated in line 225?

Line 227: 'parameterization' - meaning what? - do you mean propagation?

Line 241: Here you suddenly mention Malaise traps - any particular reason for that? Were Malaise traps the major collection method? - not mention in the methods ...

Line 252: No - a barcode study cannot say which group is the more diverse! It depends on sampling strategies and time. Beetles differ tremendously from flies and wasps because they are on average very specimen poor and rarely occur in such large numbers as do flies and wasps.

Line 257: endemism - yes, partly true, but remember that neighboring countries are also poorly investigated and the Pakistani fauna is largely shared with Iran and north to the Caucasus areas etc. The fauna in Pakistan is not so endemic as the data in this barcode study may indicate.

·

Basic reporting

I enjoyed reading this paper, which presents an impressive DNA barcode dataset on insects from Pakistan. It is written in clear and unambiguous English with citations to relevant literature. The article is well organized and figures and tables presents results in a professional manner.
The raw data is available in an open dataset in BOLD. The results are discussed appropriately in context with similar studies and present valuable new knowledge about the insect diversity in Pakistan.

The introduction gives proper background to the field, but I think the reader would appreciate a brief reminder of situations where DNA barcodes do not work so well to delimitate species (e.g. reference to Wolbachia, recent speciation, incomplete lineage sorting, etc.). Moreover, the BIN system has been found to overestimate diversity in certain insect groups such as Sciaridae and Chironomidae (e.g. https://doi.org/10.1139/gen-2018-0100, http://dx.doi.org/10.1371/journal.pone.0138993). The actual reason for this is unknown, but it is not unlikely that large effective population sizes over long evolutionary time periods could be one explanation. I do not think these facts affects the general and new results presented here, but I think it is worth mentioning in the introduction and in the discussion since it might affect the species diversity estimates for Diptera.

Experimental design

The article presents original primary research and is within the scope of the journal.
The research is performed to a high technical and ethical standard.

Please provide reference to the stand-alone version of the RESL algorithm used and where this tool can be accessed.

Validity of the findings

The findings are sound and valid and conclusions are appropriate. However, please see comment under section 1.

Additional comments

Two formulations should be changed:

Line 172: I suppose you mean the "remaining eight" orders, not the "other".
Line 200: Please reformulate because 2684 BINs are not shared with 199 other countries, but with at least one of 199 other countries.

---

## Round 0.2 · accepted · Accept

Dear Dr. Ashfaq and colleagues:

Thanks for revising your manuscript based on the concerns raised by the reviewers. I now believe that your manuscript is suitable for publication. Congratulations! I look forward to seeing this work in print, and I anticipate it being an important resource for groups studying insect diversity in Pakistan. Thanks again for choosing PeerJ to publish such important work.

Best,

-joe

·

Basic reporting

No more comments.

Experimental design

No more comments.

Validity of the findings

Noe more comments.

Additional comments

> Keywords: Change "cytochrome oxidase I" to "cytochrome c (in italics) oxidase subunit I"
> Be aware to write Wolbachia in italics (It is a genus that contains various species)